# *Cladosporium* sp. Isolate as Fungal Plant Growth Promoting Agent

**Iuliana Răut** [1], **Mariana Călin** [1], **Luiza Capră** [1], **Ana-Maria Gurban** [1,*], **Mihaela Doni** [1], **Nicoleta Radu** [1,2] and **Luiza Jecu** [1,*]

1   Biotechnology Department, National Institute for Research & Development in Chemistry and Petrochemistry—ICECHIM, Spl. Independentei 202, 060021 Bucharest, Romania; iulia_rt@yahoo.com (I.R.); marriconstantin@yahoo.com (M.C.); luizacapra@yahoo.com (L.C.); mihaela.badea@icechim.ro (M.D.); nicolbiotec@yahoo.com (N.R.)
2   Faculty of Biotechnology, University of Agronomic Sciences and Veterinary Medicine of Bucharest, 59 Mărăşti Boulevard, 011464 Bucharest, Romania
*   Correspondence: amgurban@yahoo.com (A.-M.G.); jecu.luiza@icechim.ro (L.J.); Tel.: +40-0213163063 (L.J.)

**Abstract:** *Cladosporium* species are active in protecting plants against different biotic and abiotic stresses. Since these species produced a wide range of secondary metabolites responsible for the adaptation to new habitats, plant health and performance, they are of great interest, especially for biostimulants in agriculture. *Cladosporium* sp. produces protein hydrolysates (PHs), a class of biostimulants, by cultivation on medium with keratin wastes (feathers) as carbon and energy sources. The aim of this study was to select a *Cladosporium* isolate with potential to be used as plant growth promoting agent. The characteristics of *Cladosporium* isolates as plants biostimulants were evaluated through several tests, such as: antagonism versus plants pathogens, effect on plant growth of secreted volatiles produced by isolates, secretion of hydrolytic enzymes, production of 3-indole acetic acid, zinc and phosphorous solubilization, capacity to promote tomato seedlings growth (pot experiments). *Cladosporium* isolate T2 presented positive results to all tests. Encouraging results were obtained treating tomato seedlings with PHs from isolate *Cladosporium* T2 cultured on medium supplemented with 1% (*w/w*) chicken feathers, for which growth parameters, such as stem weight, stem height, and root weight were significantly higher by 65%, 32%, and 55%, respectively, compared to those treated with water.

**Keywords:** biostimulants; *Cladosporium*; feathers; protein hydrolysates; plant growth promoting agent

## 1. Introduction

*Cladosporium* species, endophytic fungi world widely distributed, are known to be active in protecting plants against different biotic and abiotic stresses. Based on the secretion of beneficial secondary metabolites, these species improve the ability of plants to adapt to new habitats and to sustain the plant health and performance. Among the secreted metabolites, a key role is played by gibberellins, hormones responsible for plant growth-stimulating, especially in seed germination, stem elongation and leaf expansion [1–3]. Despite the colonization of internal plant tissues, *Cladosporium* species are not producing damages, and they are of great interest for specific medical effects, such as beneficial human health effects [4] and agro-industrial applications [5–8], as in the bioremediation and detergent industry [9] or in discoloration of textile dyes [10] and in the degradation of keratin containing wastes from natural environment [11,12].

Over the years, due to the increasing demand for better yield and quality of food and crops, the attention was focused on waste-derived biostimulants or raw organic material with biostimulant components [13–16]. Among biostimulants, protein hydrolysates (PHs)

from fungal cultures carried out on medium with keratin wastes as a source of carbon and energy could be efficient in plants treatment for improving crop productivity and quality.

Keratin waste is an abundant and valuable resource, and its accumulation creates serious problems for the environment, only an efficient management will reduce soil and groundwater pollution. There are many possible applications of keratin waste for the reinforcement of cement, biopolymers, and composites, as well as in many biomedical applications [17,18]. Poultry feathers from processing chicken meat represent the most part of keratin waste and have become one of the major pollutants due to the recalcitrant nature of keratin [19]. Many restrictions prevent disposal methods and it is interesting to assess the potential of waste feathers in higher added-value applications. Keratin can be hydrolyzed by chemical treatment, or by hydrothermal process with the disadvantage of producing by-products of high toxicity to humans and environment. At the same time, keratin suffers severe degradation and destruction which affect further applications. The mild conditions of microbial degradation of keratin by keratinolytic microorganisms (fungi or bacteria) are considered an eco-friendly approach producing keratinases and reducing the amount of keratin waste [20–23]. The biodegradation process is efficient, with low costs and ensures the valorization of keratin waste while maintaining the valuable nutrient composition. The secreted keratinases have various applications in cosmetic, animal feed, detergents and agriculture [24].

The ability of *Cladosporium* to degrade the extensive disulfide crosslinking of keratin polypeptides and solubilize the keratin by secreting specialized enzymes can be used to obtain plant biostimulants, and recommends *Cladosporium* as a fungal agent to promote plant growth.

In view of the above facts, this study was designed to select a *Cladosporium* isolate with potential to be used as plant growth promoting agent. For this purpose, we analyzed the following issues: (i) To reveal the secretion of volatile organic compounds and the biocontrol activity against selected phytopathogens; (ii) to detect several significant abilities for a candidate as plant growth promoting agent, namely, phosphate and zinc solubilization activities, plant hormone production such as 3-indole acetic acid and the secretion of hydrolytic enzymes; and (iii) to assess *Cladosporium* sp. isolates as fungal agent for promoting plant growth through pot experiments. This combined analysis provided important information on the characteristics of *Cladosporium* isolate, which guided its application as plant biostimulants in agriculture.

## 2. Materials and Methods

### 2.1. Microorganism

Three *Cladosporium* soil isolates belonging to Microbial Collection of ICECHIM were used in the experimental study. The stock isolates were maintained on potato-dextrose-agar medium (PDA, Scharlau) and prior to use, their purity was controlled. The composition of the PDA medium was (g·L$^{-1}$): 4, peptone; 20, glucose, 15, agar; final pH = 5.6 at 25 °C.

### 2.2. Keratin Substrate

Keratin powder was obtained as follows: chicken feathers from local poultry were cleaned, sterilized with 3% ethanol, vigorously washed and dried at 60 °C. Following disinfection, the feathers were cut into small pieces for fungal cultivation. For use in keratinase assay as substrate, the feathers were grounded several times with a Retsch ball mill until they became a fine powder [25].

### 2.3. Fungal Cultivation for Protein Hydrolysates (PHs) Preparation

Cultivation in liquid medium was performed in Erlenmeyer agitated flasks on rotary incubator shaker Heidolph Unimax 1010, at 26 ± 2 °C for 21 days. The mineral basal medium had the following composition (g·L$^{-1}$): 0.1, $KH_2PO_4$; 0.1, $CCl_2$; 0.1, $FeSO_4 \times 7H_2O$; 0.005, $ZnSO_4 \times 7H_2O$. The culture medium was supplemented with 1% feathers (processed as above, cut into small pieces), as a sole source of carbon and nitrogen and inoculated with

5 mL of each preculture. After incubation, the culture broth was filtered and supernatants were tested in experiments.

### 2.4. Antagonism versus Plants Pathogens

Cladosporium isolates were in vitro tested for antagonistic activity by dual culture technique on PDA plates [26]. Four plant pathogens purchased from Deutsche Sammlung von Mikroorganismen und Zellkulturen GmbH (DSMZ) were tested: *Rhizoctonia solani*, *Fusarium graminearum*, *Sclerotinia sclerotiorum*, and *Botrytis allii*. Briefly, PDA medium poured in sterile Petri plates was incubated with 5 mm culture disc of isolate and each pathogen. Antagonist activity was observed after incubation at $25 \pm 1$ °C, for 3–5 days. The inhibition range was determined with the formula:

$$I\ (\%) = (M\ -P)/M \times 100,$$

where $I$ = inhibition; $M$ = diameter colony for the control of pathogen; $P$ = colony diameter of pathogen in dual cultures in the presence of antagonist. Three independent experiments were carried out.

### 2.5. Evaluation of Tomato Growth Promotion by Volatiles Produced by Isolates

The tests were carried out with tomato seeds (*Solanum lycopersicum)* purchased from Agrosem (Tg-Mureş, Romania). The seeds were surface sterilized by soaking in 75% ethanol and immersion in 3% sodium hypochlorite. The fungal isolates were cultured on potato dextrose agar medium (Scharlau). The test was carried out in Magenta flasks [27], as follows: 1 mL of PDA medium was placed in each of the 12 glass bottles, and of these, only 9 bottles were inoculated with 10 μL sporal suspension containing $10^5$ ufc/mL of *Cladosporium* isolates, three bottles for each isolate, and three for control, without microorganism. The bottles were sealed with biological filters that restrict the movement of conidia, but not volatiles. Plant growth was monitored and compared with controls grown without fungal cultures. The measured growth parameters included the total height and weight of the fresh stem, the weight of the root system and the number of leaves.

### 2.6. Enzymatic Activities

(1) Chitinase activity was evaluated by qualitative method with colloidal chitin as carbon source [28]. The culture medium had the following composition ($g \cdot L^{-1}$): 4.5, colloidal chitin; 0.3, $MgSO_4X7H_2O$; 3.0, $(NH_4)_2SO_4$; 2.0, $KH_2PO_4$; 1.0, citric acid; 15.0, agar; 0.15, bromcresol purple; 200 μL, Tween 80. The Petri plates were central inoculated, as follows: (i) *liquid culture filtrate* (20 μL) of each Cladosporium isolate grown on medium with or without feathers, and, (ii) spore suspension from Cladosporium culture on PDA medium, respectively. A 1 mL spore suspension contained approx. $1 \times 10^5$ spore·mL$^{-1}$. The Petri plates were incubated at $26 \pm 2$ °C for seven days. The appearance of red-violet color was considered a positive result. Three replicates were made in all experiments.

(2) Cellulase activity was evaluated by the qualitative method with carboxymethyl-cellulose (CMC) as a cellulosic substrate [29]. The solid medium had the following composition ($g \cdot L^{-1}$): 1, yeast extract; 5, CMC; 15, agar. The Petri plates were inoculated and incubated as in the chitinase assay. After incubation, the plates were stained with Lugol's solution or Congo red (0.1%), respectively, for 15 min, at $26 \pm 2$ °C. The formation of a transparent circular area around fungal colony was considered a positive result. Three replicates were made.

(3) Keratinase activity was evaluated by the qualitative method using ball-milled feathers powder as an enzymatic substrate [25,30]. Chicken feathers were disinfected with 70% (*v/v*) ethylic alcohol, rinsed with sterile distilled water, dry overnight at 50 °C, and finally, milled in ball mill Retsch (Model MM400, Germany). The solid medium had the following composition ($g \cdot L^{-1}$): 10, feathers; 0.1, $KH_2PO_4$; 0.01, $CaCl_2$; 0.1, $FeSO_4X7H_2O$; 0.005, $ZnSO_4 \times 7H_20$; pH = 7. The Petri plates were inoculated and incubated as in chitinase

assay. Fungal growth was evaluated by measuring the colony diameter after, respectively, five and 10 days. Three replications were made.

### 2.7. Colorimetric Assay for 3-Indole Acetic Acid (IAA) Production

A colorimetric assay was used for IAA production at *Cladosporium* isolates [31]. Two compositions of culture medium were used (g·L$^{-1}$): (i) PDB medium (potato-dextrose-broth, Scharlau), as control; (ii) PDB medium supplemented with L-tryptophan (0.1 g·L$^{-1}$). The flasks inoculated with the fungal mycelium fragment were incubated for 8 days, at 26 $\pm$ 2 °C, on Heidolph Unimax 1010 (Heidolph Instruments GmBH and Co, Germany) at 130 rpm. After centrifugation at 4000 rpm, 15 min, at 4 °C, 2 mL of each collected supernatant were treated with two drops of ortho-phosphoric acid. The positive result is considered the appearance of pink color in test slants. The quantification was made using the absorbance measured at 530 nm wavelength, on BioMate$_3$ (Thermo Spectronic, USA) [32], by reference to the calibration curve drawn with commercial IAA (Sigma Aldrich). Three replications were made.

### 2.8. In Vitro Zinc Solubilization Assessment Using Plate Assay

The fungal isolate was tested for its zinc solubilizing ability using an insoluble zinc compound, as ZnO [33]. The cultivation medium had the following composition (g·L$^{-1}$): 10, dextrose; 1, $NH_4SO_4$; 0.2, KCl; 0.2, $K_2HPO_4$; $MgSO_4X7H_2O$; 1, ZnO; 15, agar. The medium was distributed into Petri plates and then inoculated with 5 mm mycelium fragment from fungal culture. The inoculated plates were incubated in the dark at 26 $\pm$ 2 °C, for seven days, in order to observe clear halo area around the colonies. The diameter of the clear zone was recorded. Three replicates were made.

### 2.9. In Vitro Phosphate Solubilization Assessment Using Plate Assay

Phosphate solubilization was qualitatively determined [34]. *Cladosporium* isolates were cultured in medium with the following composition (g·L$^{-1}$): 0.3, $MgSO_4X7H_2O$; 0.004, $MnSO_4X7H_2O$; 0.002, $FeSO_4X7H_2O$; 20, NaCl; 0.5, yeast extract; 0.1, brom cresol purple as pH indicator; 5.0, $Ca_3(PO_4)_2$; 16, agar; pH = 7.0. The Petri plates were inoculated with 5 mm mycelia fragments from fresh cultures and incubated for five days, at 26 $\pm$ 2 °C. The phosphate solubilization was confirmed by the formation of the clearing zone around the colonies. Three replications were made.

### 2.10. Nitrogen Content of Protein Hydrolysates

The supernatants were analyzed for nitrogen content according to the Kjeldhal method [35], using SR EN 15475:2009, EN 15558:2009 and EN 15558:2009 protocols. The analytical system (Behrotest S4$^®$ WD 40, Behr Labor-Technik, Dusseldorf, Germany) comprises: infrared rapid digestion equipment, process suction automatic scrubber for neutralization of $H_2SO_4$ vapors and a fully automatic steam distillation equipment. The total protein content was calculated by multiplying the protein nitrogen concentration by a factor of 6.25.

### 2.11. Test In Vivo for the Capacity to Promote Tomato Seedlings Growth (Pot Experiments)

Tomato seeds (*Solanum lycopersicum*) were planted in multicellular trays and incubated in the growth chamber under controlled conditions (Micro Clima Series TM, Labs Economic Lux chamber, Snijders, Netherlands). The parameters were set at the following values: humidity about 69%; 26 °C as the day temperature and 22 °C as the night temperature; 10,000 lux; day/night cycle as 16 h per day/8 h per night. Four multicellular trays (12 cells per tray) were used for each type of treatment and also for control. All treatments were performed with 1 mL of protein hydrolysates filtrate obtained from the culture liquid after filtration with a 0.2 μm filter, over a period of 1 month. The treatments were applied at weekly intervals. The control trays were treated with water. Additonally, mineral medium was used for the treatment of corresponding pots. After 1 month, the tomato seedlings were carefully collected from the multicellular trays and the root was cleaned with tap

water for several times, in order to analyze the planned growth parameters. The growth parameters of the treated tomato seedlings were compared with the untreated ones [36].

*2.12. Statistical Analysis*

The data obtained in each experimental variant were analyzed with GraphPad Prism 5.0 software. All parameters (Standard Error, p value) were calculated using a confidence interval of 95%. Results are presented as standard error bars and respectively as "*p*" values. In the case of p parameter were used the following significances and notations: $p < 0.05$ = insignificant differences; notation: * for $0.05 < p < 0.01$ = significant differences between the experimental variant and Control; notation: ** for $0.01 < p < 0.005$ = distinct significant differences between the experimental variant and Control; notation: *** for $p < 0.005$ = highly significant differences between the experimental variant and Control.

## 3. Results

*3.1. Antagonism versus Plant Pathogens*

The tests were carried out in Petri plates using several plant pathogens such as: *F. graminearum, S. sclerotiorum, B. allii*, and *R. solani*, the results being presented in Figure 1 and respectively in Table 1.

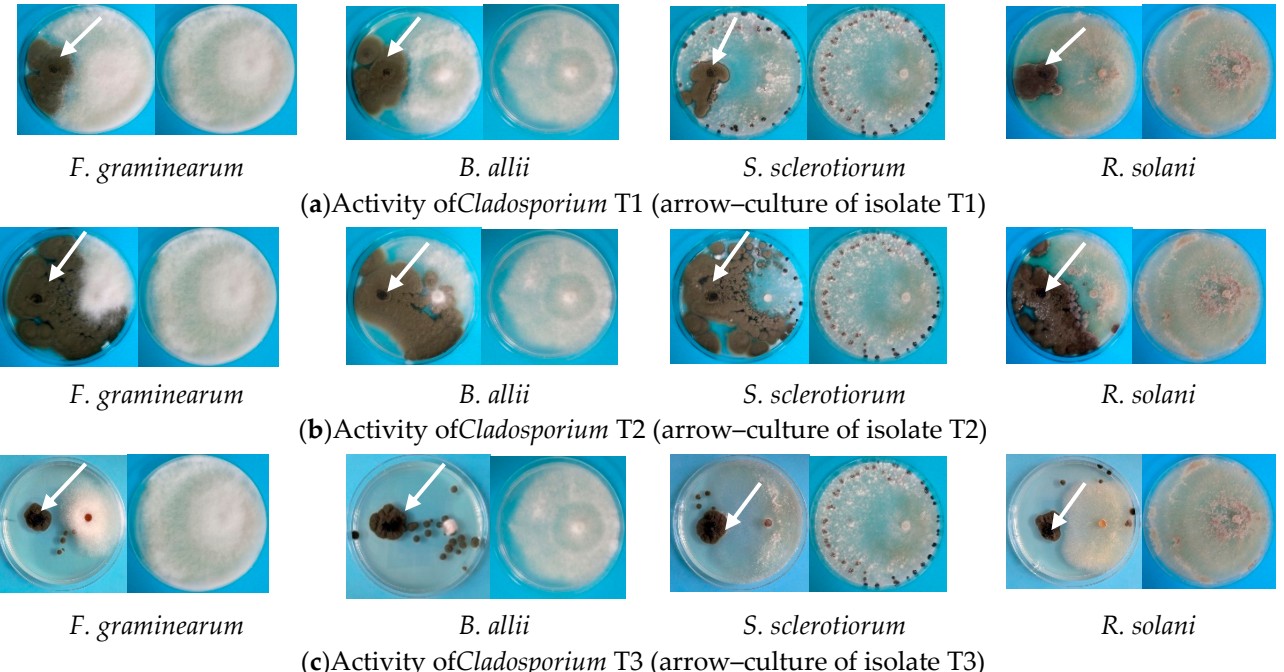

F. graminearum  B. allii  S. sclerotiorum  R. solani
(**a**)Activity of*Cladosporium* T1 (arrow–culture of isolate T1)

F. graminearum  B. allii  S. sclerotiorum  R. solani
(**b**)Activity of*Cladosporium* T2 (arrow–culture of isolate T2)

F. graminearum  B. allii  S. sclerotiorum  R. solani
(**c**)Activity of*Cladosporium* T3 (arrow–culture of isolate T3)

**Figure 1.** Antagonism of *Cladosporium* isolates *versus* plant pathogens. Pure fungal colonies of pathogens are presented as growth control.

**Table 1.** Inhibition of pathogens by *Cladosporium* isolates.

| Antagonistic Fungal Isolate | Inhibition * (%) | | | |
|---|---|---|---|---|
| | *F. graminearum* | *B. allii* | *S. sclerotiorum* | *R. solani* |
| *Cladosporium* sp. T1 | 35.6 ± 0.5 | 37.1 ± 0.4 | 40.8 ± 0.9 | 35.5 ± 0.5 |
| *Cladosporium* sp. T2 | **57.4 ± 0.5** | 86.4 ± 0.5 | **62.6 ± 0.5** | **56.9 ± 0.9** |
| *Cladosporium* sp. T3 | 47.1 ± 0.4 | 58.3 ± 0.6 | 29.2 ± 0.4 | 41.7 ± 0.5 |

* Values are the average of three independent experiments ± standard deviations.

From all tested strains, *Cladosporium* T2 isolate was the most active against tested pathogens, *B. allii,* being most sensitive to antagonist isolate, with an inhibition rate of 86.4%. This is a valuable information since *Botrytis allii* affects onions, garlic, and leeks, causing economical post-harvest damages, decreasing the storage durability and market value.

### 3.2. Effects of Volatiles Secreted by Cladosporium Isolates

Tomato seeds were exposed to volatiles produced by *Cladosporium* isolates and their effect upon germination process is presented in Figure 2. By emitting volatile blends, *Cladosporium* isolates mediated the growth of tomato plants as compared to control, where the seeds were treated with water. Thus, a much more vigorous growth was observed, with more robust stems and leaves, compared to the control seedlings not exposed to volatile compounds. For measuring the plant growth promotion under in vitro conditions, 21 days exposure duration have been used, being comparable with other similar studies carried out with tobacco plants [37].

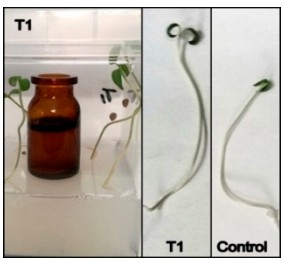 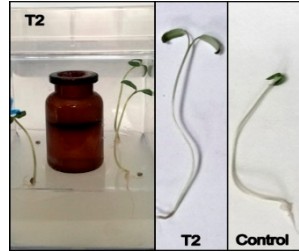 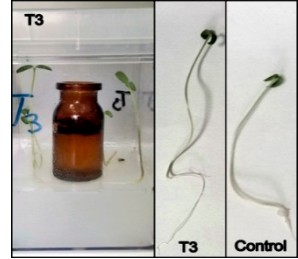 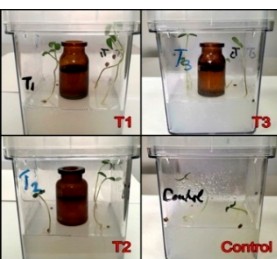

Exposure toT1 isolate   Exposure toT2 isolate   Exposure toT3 isolate   Exposure toall isolates

**Figure 2.** Growth promotion of tomato seedling by exposure to volatiles secreted by *Cladosporium* isolates. The photographs were taken 21 days after seed sowing.

After 21 days of exposure to volatiles secreted by *Cladosporium*, the individual plants presented certain increases as compared with controls. Thus, an equal increase of 50% for total plant weight at all fungal isolates was recorded compared to the control values. The percentage increase for plant height compared to control follows the descendent order: 28.5% for isolate T2, >27% for isolate T3 and, respectively, >25% at isolate T1. The highest values for plant height were obtained for T1 and T2 (7.5 cm; increase of 26% compared to control), followed by T3 (6.8 cm; increase of 19%). Regarding the length of root system, the best results were found at isolate T2 (3.5 cm), followed by isolate T3 (3.2 cm), and isolate T1 (2.5 cm), compared with the control values (2.0 cm). No significant difference between isolates was observed regarding the number of leaves (Table 2).

**Table 2.** Plant growth parameters after volatiles exposure.

| Fungal Isolate | Growth Parameters * | | | | |
|---|---|---|---|---|---|
| | Total Weight (g) | Total Height (cm) | Plant Height (cm) | Root System Length (cm) | Number of Leaves |
| *Cladosporium* sp. T1 | 0.06 ± 0.01 | 10.0 ± 0.3 | 7.5 ± 0.2 | 2.5 ± 0.2 | Two leaves of medium size |
| *Cladosporium* sp. T2 | 0.06 ± 0.01 | **10.5 ± 0.2** | **7.5 ± 0.4** | **3.5 ± 0.3** | Two leaves of medium size |
| *Cladosporium* sp. T3 | 0.06 ± 0.01 | 10.3 ± 0.3 | 6.8 ± 0.4 | 3.2 ± 0.2 | Two leaves of medium size |
| Control | 0.03 ± 0.01 | 7.5 ± 0.2 | 5.5 ± 0.2 | 2.0 ± 0.3 | Two leaves of small size |

* Values are the average of three independent experiments ± standard deviations.

### 3.3. Secretion of Hydrolytic Enzymes

The presence of hydrolytic enzymes was highlighted in plate assay by inoculating the solid medium with mycelia fragments from isolates or filtrates from fungal cultivations in liquid media, with or without feathers (Figure 3). All enzymatic activities exhibited by isolate T2 were significantly higher compared to isolates T1 and T3, especially for cellulase activities where clearance zone (halo) after flooding with Lugol covered almost the entire plate (Figure 3c).

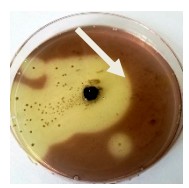
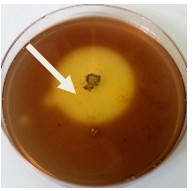

*Cladosporium* sp. T1          *Cladosporium* sp. T2          *Cladosporium* sp. T3

(**a**)Keratinase plate assay. Zone of clearance (halo) after flooding with iodine solution (Lugol).

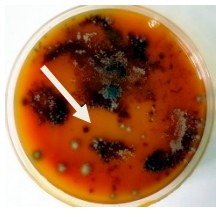
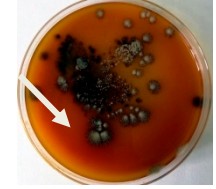
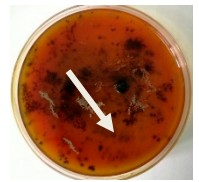

*Cladosporium* sp. T1          *Cladosporium* sp. T2          *Cladosporium* sp. T3

(**b**)Chitinase plate assay**.** Changing the color of culture medium from yellow to reddish brown, around the mycelium

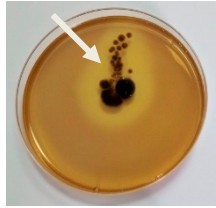
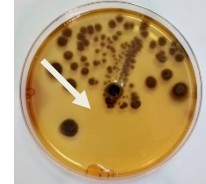
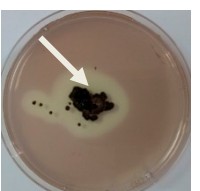

*Cladosporium* sp. T1          *Cladosporium* sp. T2          *Cladosporium* sp. T3

(**c**)Cellulases plate assay.Zone of clearance (halo) after flooding with iodine solution (Lugol)

**Figure 3.** Secretion of hydrolytic enzymes through plate assay inoculated with mycelia fragments at *Cladosporium* isolates (after 7 days of incubation).

*Cladosporium* isolate T2 grown on feathers medium exhibited the highest values of keratinase, chitinase, and cellulase, followed by isolate T3 (Table 3.).

**Table 3.** Colony diameter in plate assay inoculated with culture filtrates for detection of hydrolytic enzymes.

| Culture Filtrate from Isolates | Colony Diameter (cm) * | | |
|---|---|---|---|
| | *Keratinase* **Activity** | *Chitinase* **Activity** | *Cellulase* **Activity** |
| *Cladosporium* sp. T1 (medium without feathers) | 0 | 0 | $0.6 \pm 0.05$ mm |
| *Cladosporium* sp. T1 (medium with feathers) | $2.6 \pm 0.05$ | $2.2 \pm 0.1$ | $2.2 \pm 0.05$ |
| *Cladosporium* sp. T2 (medium without feathers) | $0.7 \pm 0.05$ | $0.7 \pm 0.1$ | $0.6 \pm 0.05$ |

**Table 3.** *Cont.*

| Culture Filtrate from Isolates | Colony Diameter (cm) * | | |
|---|---|---|---|
| | *Keratinase* Activity | *Chitinase* Activity | *Cellulase* Activity |
| *Cladosporium* sp. T2 (medium with feathers) | **4.3 ± 0.05** | **3.4 ± 0.09** | **3.1 ± 0.08** |
| *Cladosporium* sp. T3 (medium without feathers) | 0.6 ± 0.06 | 0.8 ± 0.05 | 0.8 ± 0.08 |
| *Cladosporium* sp. T3 (medium with feathers) | 3.2 ± 0.00 | 3.2 ± 0.05 | 2.4 ± 0.09 |

* Values are the average of three independent experiments ± standard deviations.

No keratinase and chitinase activities were obtained from *Cladosporium* isolate T1. As expected, the enzymatic activities exhibited by fungal isolates were higher in the medium containing feathers, considered a good source of carbon and energy. No clear zones were formed in tests carried out with filtrates from minimal culture medium and non-inoculated medium, as control.

*3.4. IAA Production*

The production of IAA was determined using a colorimetric method. Figure 4 shows that only *Cladosporium* isolate T1 failed to produce IAA, while the test results were positive for T2 and T3 isolates, highlighting the apperance of pink color.

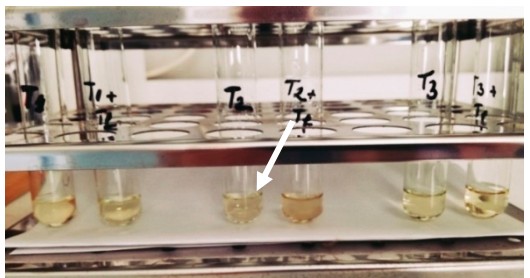

(**a**) Colorimetric test for IAA production

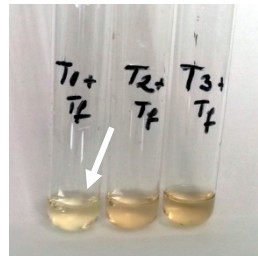

(**b**) PDB medium + L-tryptophan

**Figure 4.** Images of test for IAA production at fungal isolates (T1, T2, T3—supernatants from fungal cultivation on PDB medium; T1 + Tf, T2 + Tf, T3 + Tf—supernatants from fungal cultivation on PDB medium supplemented with L-tryptophan; arrow—pink color).

*Cladosporium* isolate T2 was about 62% more active in IAA production on PDB medium, and 22% on PDB + trytophan medium, respectively, compared to T3, while T1 isolate showed very low activity closer to control values (Table 4).

**Table 4.** Production of IAA by fungal isolates on culture media.

| Strain | DO*$_{530\,nm}$ (PDB Medium) | DO*$_{530\,nm}$ (PDB + L-Tryptophan) |
|---|---|---|
| *Cladosporium* T1 | 0.014 ± 0.001 | 0.024 ± 0.001 |
| *Cladosporium* T2 | **0.076 ± 0.002** | **0.125 ± 0.012** |
| *Cladosporium* T3 | 0.047 ± 0.001 | 0.102 ± 0.002 |
| Control (culture medium) | 0.010 ±0.002 | 0.014 ± 0.001 |

* Values are the average of three independent experiments ± standard deviations.

*3.5. Capacity of Zinc and Phosphorus Solubilising*

The special features as a promoting agent for plant growth are the ability to solubilize zinc and phosphorus (Figure 5). The tests were performed on Petri plates on a specific

media composition, ZnO and calcium phosphate, respectively. The activity for zinc solubilization of *Cladosporium* T2 isolate was higher compared to T1 and T3 isolates (Figure 5a). Regarding phosphorus solubilization, the differences between isolates were not significant, the images indicating almost the same zone of decoloration (Figure 5b).

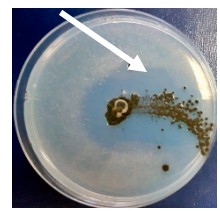
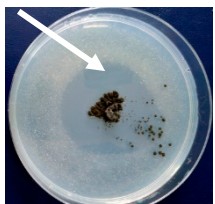

*Cladosporium* sp. T1      *Cladosporium* sp.T2      *Cladosporium* sp.T3

(**a**)Zn (as 0.1% ZnO) solubilization. Appearance of clear zone (halo) around mycelium, as Zn solubilization

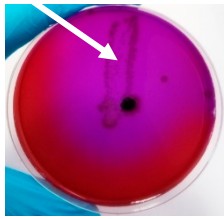
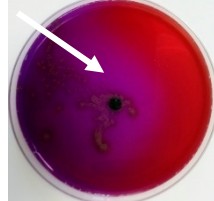
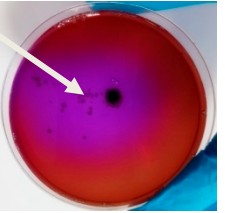

*Cladosporium* sp. T1      *Cladosporium* sp.T2      *Cladosporium* sp.T3

(**b**)Phosphorus (as Ca3 (PO4)2 solubilization. Large zone of decolorization in culture me-dium (changing the color in purple from red, around fungal colony.

**Figure 5.** Images of zinc and phosphorus solubilization at *Cladosporium* isolates (images after seven days of incubation).

### 3.6. Nitrogen Content

Useful data for the characterization of fungal protein hydrolysates were obtained from the analysis of organic and inorganic nitrogen content. Cultivation of Cladosporium T2 isolate on feathers medium led to protein hydrolysates with the highest level of ammonium and total nitrogen, $211 \times 10^{-3}$ mg/mg, and $569 \times 10^{-3}$ mg/mg, respectively, compared to other working options (Table 5.). Generally, the total nitrogen content of PHs obtained in medium with and without keratin source decreased in the following order: isolate T2 > isolate T1 > isolate T3, except for ammonium nitrogen in medium without feathers, where the value from T3 exceeded those obtained from T1.

**Table 5.** Nitrogen content of protein hydrolysates obtained from cultivation of *Cladosporium* isolates in experimental conditions (*).

| Sample | Total Nitrogen * (Kjeldhal) (mg/mg) | | Ammonium Nitrogen * (mg/mg) | | Protein Nitrogen * (mg/mg) | |
|---|---|---|---|---|---|---|
| | Medium no Feathers | Medium with Feathers | Medium no Feathers | Medium with Feathers | Medium no Feathers | Medium with Feathers |
| *Cladosporium* T1 isolate | $127 \times 10^{-3}$ | $137 \times 10^{-3}$ | $61 \times 10^{-3}$ | $87 \times 10^{-3}$ | $66 \times 10^{-3}$ | $50 \times 10^{-3}$ |
| *Cladosporium* T2 isolate | $90 \times 10^{-3}$ | $\mathbf{569 \times 10^{-3}}$ | $93 \times 10^{-3}$ | $\mathbf{211 \times 10^{-3}}$ | $55 \times 10^{-3}$ | $\mathbf{358 \times 10^{-3}}$ |
| *Cladosporium* T3 isolate | $102 \times 10^{-3}$ | $120 \times 10^{-3}$ | $88 \times 10^{-3}$ | $87 \times 10^{-3}$ | $33 \times 10^{-3}$ | nd |
| Minimal basal medium-Control | $80 \times 10^{-3}$ | $96 \times 10^{-3}$ | $60 \times 10^{-3}$ | $84 \times 10^{-3}$ | $20 \times 10^{-3}$ | $12 \times 10^{-3}$ |

(*) Total nitrogen–Kjeldhal method; ammonium nitrogen calculated according to standard SR EN 15475:2009; protein nitrogen–calculated as total nitrogen minus ammonium nitrogen.

### 3.7. Pot Experiments with Tomato Seedlings Treated with Protein Hydrolysates from Cladosporium Cultures

The growth parameters as plants height, shoots length and root weight of the treated tomato seedlings were compared with the untreated ones after 1 month of incubation in the growth chamber (Figure 6).

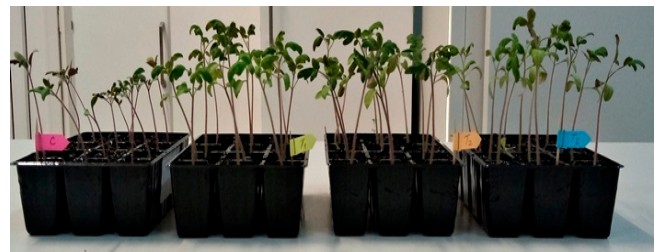

(**a**) Plants exposed to PHs from isolates cultured on medium *without feathers*

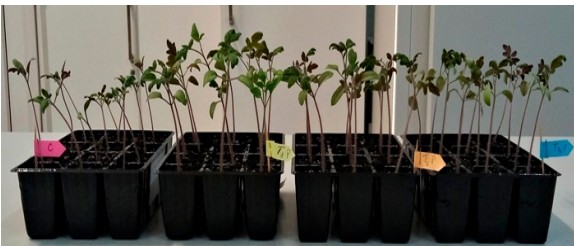

(**b**) Plants exposed to PHs from isolates cultured on medium *with feathers*

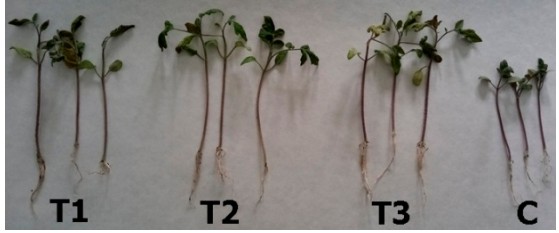

(**c**) Plants collected after 1 month of treatment with PHs from isolates grown on medium *without feathers*

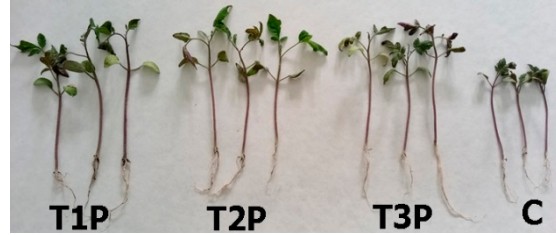

(**d**) Plants collected after 1 month of treatment with PHs from isolates grown on medium *with feathers*

**Figure 6.** Images of tomato plants treated with protein hydrolysates from *Cladosporium* isolates (fungal isolates T1, T2, and T3 cultured on medium without feathers; fungal isolates T1P, T2P and T3P cultured on medium with feathers; C—Control, treatment with tap water).

In pot experiments, all *Cladosporium* isolates presented a certain positive action on tomato seedlings by improving the growth parameters. Visual observations showed that tomato plants used as control, treated with water or minimal medium, are smaller compared to those treated with PHs (Table 6). Thus, the data demonstrate the performance of applying treatments with PHs compared to watering, by increasing the values of growth parameters (e.g., for stem height, 28% at T1, 32% at T2, and 30% at T3; for stem weight, 52% at T1, 65% at T2, and 55% at T3; for root weight, 55% for all isolates). As expected, the supplementation of culture medium with feathers leads to a high nitrogen content in the fungal culture filtrates, which brings benefits to plants growth, in our case, especially to rooted plant. For each fungal isolate, the values of the growth parameters were to a certain extent higher than those from culture media without feathers (e.g., for stem height, 5% at T1, 7% at T2, 9.4% at T3; for stem weight, 0 at T1, 15% at T2, 13.7% at T3; for root weight, 33% at T1, 22% at T2, and 22% at T3). It is important to highlight that the effects are more visible at root weight, the percentages are higher compared to other growth parameters. As a final conclusion of pot experiment, tomato plants treated with PHs from *Cladosporium* T2 isolate cultured on medium supplemented with feathers developed better.

**Table 6.** Effects on tomato growth exposed to treatment with protein hydrolysates from *Cladosporium* isolates cultured on medium with or without feathers.

| Protein Hydrolysates Obtained from *Cladosporium* Cultures Grown on Different Media | Stem Height * (cm) | Stem Weight * (g) | Root Weight * (g) |
|---|---|---|---|
| Isolate T1 (medium without feathers) | 8.48 ± 0.9 | 0.49 ± 0.09 | 0.06 ± 0.01 |
| Isolate T1P (medium with feathers) | 8.88 ± 0.9 | 0.48 ± 0.09 | 0.09 ± 0.02 |
| Isolate T2 (medium without feathers) | 8.62 ± 0.9 | 0.55± 0.08 | 0.07 ± 0.01 |
| Isolate T2P (medium with feathers) | **9.34 ± 0.7** | **0.65 ± 0.1** | **0.09 ± 0.03** |
| Isolate T3 (medium without feathers) | 8.33 ± 0.7 | 0.44 ± 0.08 | 0.07 ± 0.01 |
| Isolate T3P (medium with feathers) | 9.23 ± 0.9 | 0.51 ± 0.08 | 0.09 ± 0.03 |
| Minimal medium (MM) | 6.73 ± 0.8 | 0.38 ± 0.07 | 0.06 ± 0.02 |
| Control (water) | 6.37 ± 0.9 | 0.23 ± 0.07 | 0.04 ± 0.02 |

* Values are the average of measurements of 12 plants randomly selected from multicellular trays ± standard deviations.

### 3.8. Statistic Analysis

The treatment means of tomato seedlings performed in pot experiments were compared with GraphPadPrism 5.0 software. This statistical analysis provided important information about characteristics of *Cladosporium* isolates. Figure 7 provides that good results were obtained in tomato seedlings treatment with protein hydrolysates from *Cladosporium* T2 isolate (T2P sample notation) cultured on medium supplemented with 1% feathers, being statistically significant.

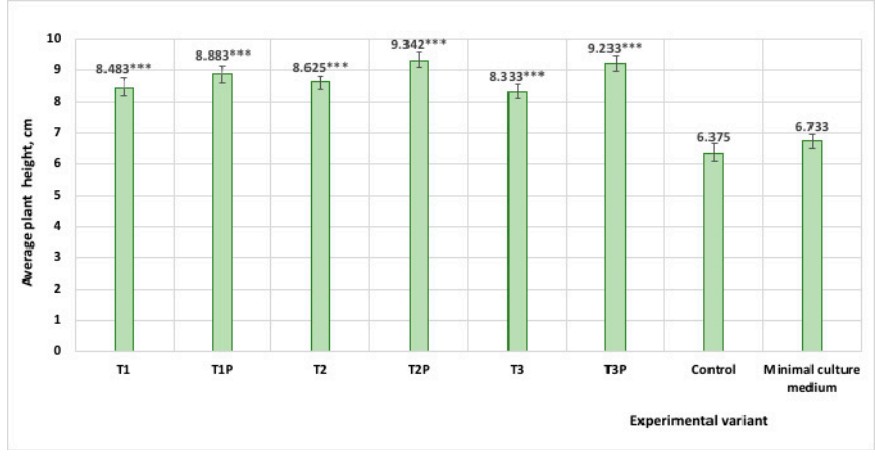

(**a**) The influence of experimental variants on stems height

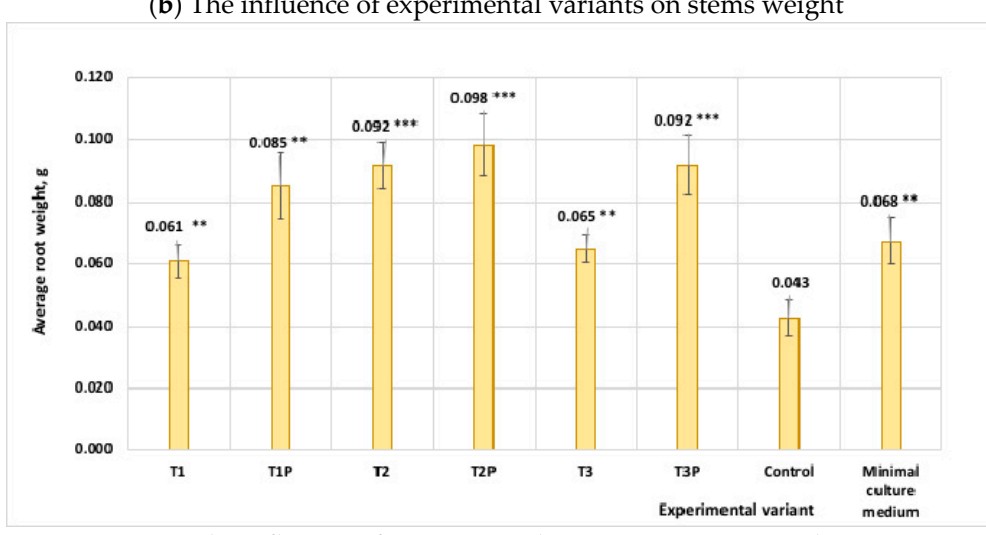

(**b**) The influence of experimental variants on stems weight

(**c**) The influence of experimental variants on roots weight

**Figure 7.** The influence of experimental variants on plants growth parameters. Measurements performed after 1 month of plants treatments with protein hydrolysates from *Cladosporium* isolates cultured on medium with or without feathers. * For $0.05 < p < 0.01$ = significant differences between the experimental variant and Control; notation: ** for $0.01 < p < 0.005$ = distinct significant differences between the experimental variant and Control; notation: *** for $p < 0.005$ = highly sig-nificant differences between the experimental variant and Control.

## 4. Discussion

Since it is particularly desirable to assess the biocontrol ability of isolates, several tests were performed using in vitro dual culture test, the most common test used for the preliminary screening of biological control agents. Our *Cladosporium* T2 isolate was able to produce a relatively good percentage of inhibition, decreasing in the following order: *B. allii* (86.4%) > *S. sclerotiorum* (62.6%) > *F. graminearum* (57.4) > *R. solani* (56.9%), which is a positive feature for future applications (Figure 1, Table 1). The studied literature shows that the antagonistic activity of species from *Cladosporium* genus has been described in many reports. For instance, *Cladosporium cladosporoides* was efficient against *Puccinia horiana*, a microcyclic rust that affects many *Chrysanthemum* species, which recommends it for an integrated management scheme applied in horticulture, especially to chrysanthemum cultivation [38]. Additionally, *Cladosporium* has been found to inhibit the growth of *Colletotrichum acutatum* by producing black spot of strawberry, or *Colletotrichum fragariae* that infects strawberries, and *Colletotrichum gloeosporioides* affecting many tropical fruits [39]. The antagonistic capability of *C. tenuissimum* was confirmed by inhibiting the germination of propagules of other rust fungi, as *Peridermium pini* and *Cronartium flaccidum* [40], and, also the germination and mycelial growth of several fungi like, *Alternaria alternata*, *Botrytis cinerea*, *Mucor sp.*, *Rhizoctonia solani* [41]. The bioactive substances produced by *Cladosporium cladosporioides* are used to control the *Aspergillus ochraceus*, *A. niger*, *Fusarium* sp., *Penicillium sp* [42] which cause major damage to coffee crops. Furthermore, *C. cladosporioides* has been shown to be a potent entomopathogenic fungus and therefore a potential candidate for biocontrol of insect pests [43].

The performance when using fungal strains as a biological control is connected to different levels of expression of hydrolytic enzymes, as keratinases, cellulases and chitinases. In some cases, hydrolytic enzymes of beneficial microorganisms may act against the fungal cell walls of pathogens. Keratinases are proteolytic enzymes that can catalyze the cleavage and hydrolysis of the highly stable and fibrous proteins: keratins. Cellulases are a complex group of three major enzymes (exoglucanases, endoglucanases, and β-glucosidase), playing an important role in the hydrolysis of lignocellulosic materials. Chitinases are glycosyl hydrolases which degrade *β*-1, 4-linked insoluble linear polymers of *N*-acetylglucosamine (chitin) directly to low molecular weight chitooligomers. It is considered that fungal chitinolytic enzymes are demonstrated to be the most effective agents as different biological control of plant infections or diseases. In our experiments. *Cladosporium* isolate T2 was the most active in producing hydrolytic enzymes, followed in decreasing order by T3 isolate, and the last displaying T1 isolate. The characterization of hydrolytic enzymes could be considered a useful tool for the selection, together with other parameters, of the best isolates for biological purposes. This is another characteristic that supports the definition of T2 isolate as a good candidate as plant biostimulant. Similar behavior has been reported for another *Cladosporium* strains. Thus, it was found that a *Cladosporium* isolated from garden soil secretes a high level of hydrolytic enzymes (lipases, proteases and DNAses), useful for the formulation of a detergent [44]. Several endophytic of *Cladosporium* genus isolated from different medicinal plants have been evaluated for enzymatic activities such as amylase, protease, cellulose, and lipase, showing a high potential for clinical microbiology and therapeutic applications [45].

Many studies highlight the important antifungal role of VOCs compounds produced by fungal species. The compounds belong to different microbial chemical classes including acids, alcohols, aldehydes, aromatics, esters, hetero-cycles, ketones, terpenes, thiols, the composition being variable in relation with fungal type and environment factors. These compounds are involved in the regulation of symbiotic associations and in the distribution of saprophytic and mycorrhizal organisms due to their features, as low molecular weight, high vapor pressure, low boiling point and a lipophilic moiety [46,47]. Also, the secreted VOCs are important as a defense weapon against pathogens, inhibiting or reducing their activities, penetrating through soil pores [48]. Volatiles emitted by *Cladosporium cladosporioides* CL-1 have significantly improved the tobacco growth [49]. The test carried out with

*Cladosporium* isolates in a Magenta vessel indicated several differences between isolates, *Cladosporium* T2 isolate determined the highest values of plants growth parameters due to volatile substances (Figure 2). The volatile substances from isolate T2 positively affect the plant growth indicating a useful feature of fungal isolate as plant promoting agent.

In the next experiment was evaluated the production of phytohormones, namely, 3-indole acetic acid, which has a significant role in plant growth and development. IAA is normally produced by several bacterial and fungal species, regulating the physiological response and gene expression in these organisms. Recent studies demonstrate the involvement of IAA in plant–pathogen interactions, such as pathogenesis and defense mechanisms [50–52]. Generally, the culture medium to produce IAA is supplemented with L-tryptophan as a precursor, meanwhile, there are fungal isolates that produced IAA in the absence of exogenous tryptophan. The positive result for our isolates T2 and T3 is considered profitable for the environment and for field crops, due to the presence of tryptophan in plant exudates that are associated with endophytic *Cladosporium*.

There are many studies showing that the ability of endophytic fungi to solubilize phosphate and zinc contributes decisively to their use as agents to promote plant growth [53,54]. Zinc is an essential micronutrient for microorganism and plants, and its deficiency in soil affects significant crop yields and quality in agriculture, retarding shoot growth, decreases leaf size, modifying the susceptibility to biotic and abiotic stresses, decreasing water uptake and transport [55]. Among our *Cladosporium* isolates, qualitative tests for zinc solubilizing showed that T2 isolate was the most active as compared to T1 and T3 isolates (Figure 5a).

Phosphate solubilization by microorganisms is a process that contributes to the growth and development of plants. Generally, the inorganic phosphorus in soil is present as crystalline unstructured calcium, aluminum, and iron phosphates. Microorganisms produce organic acids that lower the pH of environment, facilitating the exchange of cations (iron, calcium, aluminum) from insoluble phosphates to potassium or sodium, which are phosphates soluble salts [56]. Our fungal isolates were all active, with qualitative test image indicating almost the same decolorization zone for all of them (Figure 5b).

A special attention was directed to the nitrogen content of PHs used in the treatment of tomato seedlings. Nitrogen is an essential macro element, acting as key catalyst to support photosynthesis and other important biochemical reactions required for healthy plant growth. As we expected, supplementing the fungal culture medium with feathers increased the amount of nitrogen in PHs intended for plants treatment. As a consequence, PHs from *Cladosporium* T2 isolate grown on medium with 1% feathers had the highest amount of total nitrogen, and ammonium nitrogen, compared to other isolates (Table 5).

In pot experiments, treatment with PHs from *Cladosporium* isolates have beneficial effects on growth and development of plants, compared to those treated with water or minimal medium (Figure 6, Table 6). Encouraging results have been obtained in pot experiments treating tomato seedlings with protein hydrolysates obtained from *Cladosporium* T2 isolate cultured on medium supplemented with 1% feathers. Similar results have been reported, according to which *Cladosporium* strains are good agents for stimulating plant growth [27,38,49].

The combined analysis of the results from all tests provided important information about characteristics of *Cladosporium* isolate T2, which guided its application as plant biostimulants in agriculture after supplementary investigations.

## 5. Conclusions

Until few years ago, fungal endophytes are unjustly neglected although they present beneficial activities for agriculture and horticulture, being necessary a recognition and re-evaluation of their potential [57–59].

For this purpose, three *Cladosporium* isolates were analyzed for the potential to be used as plant growth promoting agents. The studied characteristics, such as secretion of volatile organic compounds, the biocontrol against several aggressive phytopathogens, phosphate and zinc solubilization activities, the production of plant hormones, the secretion

of hydrolytic enzymes and the effects on plants growth in pot experiments revealed that *Cladosporium* isolate T2 is a promising biostimulating agent with beneficial effect on crop cultures.

Our findings indicate that application of protein hydrolysates from *Cladosporium* isolate can serve as a promising approach for sustainable agriculture, as well as for the recovery of keratin wastes, abundant and cheap nitrogen-rich sources. Further studies are needed to optimize the *Cladosporium* cultivation on feathers based medium for higher nitrogen levels in protein hydrolysates, extension to field experiments, and, a very important issue, to evaluate the adverse effects (if any) on other crops, in humans or animals.

**Author Contributions:** L.J., I.R. and M.C. conceived and designed the experiments; M.C., I.R., L.C. and A.-M.G. performed the experiments; L.J., M.D. and N.R. analyzed the data; L.J., I.R. and M.C. wrote the paper; L.J., M.D. and N.R. reviewed and edited. All authors have read and agreed to the published version of the manuscript.

**Funding:** The work on this paper was supported by the Government of Romania, Ministry of Research, Innovation and Digitalization, project PN.19.23.01.01/2019.

**Acknowledgments:** The authors thank to Ministry of Research, Innovation and Digitalization of Romania, projects MANUNET-NITRISENS no 216/2020 and PN-III-P2-2.1-PED-2019-0991/PED 392/2020.

**Conflicts of Interest:** The authors declare no conflict of interest.

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
