# Peer review of "Cladosporium sp. Isolate as Fungal Plant Growth Promoting Agent"

_agronomy, doi:10.3390/agronomy11020392_

Round 1

Reviewer 1 Report

Introduction:

I suggest to the authors to increase the description of the importance of waste chicken utilization (line 50), adding some literature references on the argument. In particular, now, the sentence seems to be located there without any sense. I recommend to tie it better to the speech (connect keratine to chicken waste because the reader can be lost).

Materials and Methods:

2.1 The Cladosporium strains were previously characterize (philogenetically). These strains are in a collection. How were they characterize?

line 68. How PDA medium is composed? please, be precise.

line 84, line 130, line 407 color.

line 85-85. Do you know the strains or only species of those plant pathogens?

please uniformate the sentences time along the manuscript.

be more precise in figures lengend (abbreviations) and uniformate them (stile)

Figure 7 has a very poor quality, it is impossible to read and understand what is represented! 

Discussion:

the authors reported again the results observed in Figures but this section must be dedicated to the discussion of their own results on the base of previous literature.

They should revise and redivide results and discussion session.

Author Response

Answers to Reviewer 1

We greatly appreciate your remarks on our work and we would like to express our thanks. We made all the requested modifications according to Reviewer Comments.

Introduction:

I suggest to the authors to increase the description of the importance of waste chicken utilization (line 50), adding some literature references on the argument. In particular, now, the sentence seems to be located there without any sense. I recommend to tie it better to the speech (connect keratine to chicken waste because the reader can be lost).

We introduced discussions and 7 new references (17-24) on the use and valorization of poultry  feathers a valuable and abundant resource for various applications (lines 50-64).

Materials and Methods:

2.1 The Cladosporium strains were previously characterize (philogenetically). These strains are in a collection. How were they characterize?

The three fungal species were isolated from soil, placed on Petri plates and cultured on PDA medium (potato-dextrose-agar with antibiotics. The plates were incubated at 26 o C and periodically transferred to obtain pure cultures. The morphological characteristics of our pure fungal cultures were analyzed macroscopic (color, size, shape visible etc) and also microscopically using a lactophenol cotton blue-stained slide mounted with a small portion of the mycelium.  All three strains belong to Microbial Collection of our institute.

In conclusion, according to taxonomic literature for fungal species using databases, where there are fully description and illustrations of fungal species our fungal isolates were identified to belong to Cladosporium genus [Deacon J. W. Fungal Biology, 4th edition, ISBN-13: 978-1-4051-3066-0 (pbk. : alk. paper), ISBN-10: 1-4051-3066-0 (pbk. : alk. paper); Ellis, M.B., Dematiaceous hyphomycetes. CAB, 1971.Commonwealth Mycological Institute, Kew, Surrey, England, Surrey, 608 p.; Von Arx , J.A. The genera of fungi sporulating in pure culture, 1981, 3rd, ed. Cramer, J., Vaduz; https://drfungus.org/knowledge-base-category/fungi-descriptions/. Accessed on 2019; Mycology Online, https://mycology.adelaide.edu.au ].

The characteristics of our isolates are according to these data bases. Our isolated strains formed on the culture media colony with moderate growth rate and with a velvety to powdery texture. Colony averse is dark olive green to black and reverse olivaceous-black. On microscopic examination it was observed septate hyphae, straight conidiophores less distinct from the vegetative hyphae and conidia formed in branched chains that unfold quickly, with smooth cell wall and a dark and distinct hilum.

There are many reports on primary fungal identification based on morphological characteristics [Lima RF and Borba 2001. Viability, morphological characteristics and dimorphic ability of fungi preserved by different methods. Rev Iberoam Mycol., 2001, 18: 191-196; Gaddeyya G, Niharika PS, Bharathi P and Kumar PKR. 2012. Isolation and identification of soil mycoflora in different crop fields at Salur Mandal. AdvAppl Sci Res., 3:2020-2026; Fazio et al., Fungal deterioration of a Jesuit South American polychrome wood sculpture, Internat Biodeter Biodegrad, 2010, 64, 694-701].

It is important to mention that, this primary identification will be followed by identification using molecular biology applied to Cladosporium isolate T2 selected for potential as biostimulant agent for plant growth and development. This analysis will be performed in another institution since we have not access in our laboratory to this technique.

line 68. How PDA medium is composed? please, be precise.

Line 85-86. We added the composition of PDA medium.

line 84, line 130, line 407 color.

We corrected.

line 85-85. Do you know the strains or only species of those plant pathogens?

Line 142-143. All plants pathogens Rhizoctonia solani, Fusarium graminearum, Sclerotinia sclerotiorum and Botrytis allii were purchased from the well known Microbial Collection from Germany (Deutsche Sammlung von Mikroorganismen und Zellkulturen GmbH, DSMZ). These very aggressive pathogens producing damages to various crops and are usually present in biocontrol tests. The pathogens are taxonomic identified by DSMZ specialists.

please uniformate the sentences time along the manuscript.

We made also several modifications to improve the scientific level and English language of manuscript.

be more precise in figures lengend (abbreviations) and uniformate them (stile)

We made modifications in all figures legends.

Figure 7 has a very poor quality, it is impossible to read and understand what is represented! 

We removed the unclear photos and inserted better ones.

Discussion:

the authors reported again the results observed in Figures but this section must be dedicated to the discussion of their own results on the base of previous literature.

They should revise and redivide results and discussion session.

 We have revised results and discussions sessions to avoid repetitions (e.g. lines 434, 571, 618, 714, 730, 737).

Thank you for support and collaboration

Yours sincerely

Luiza Jecu

17.02.2021

Reviewer 2 Report

The manuscript by Iuliana Raut and co-autors titled “Cladosporium sp. isolate as fungal plant growth promoting agent” is very interesting, deals with the important information about characteristics of Cladosporium isolate, and its application as plant biostimulants in agriculture.  For this reason, the issues raised by the authors are cognitively valuable. Methods of the experiment are completely described.Therefore, the results are valuable and worth publishing. However, the publication requires the introduction of necessary corrections and supplementations indicated below for the Authors' consideration:

  • please describe in detail the conditions in growth chamber (e.g. temperature, humidity and photoperiod),
  • You specify that you are using the notation: *0.01 < p<0.001 = distinct significant differences between the experimental variant and Control; notation: ** p < 0.001 = highly significant differences between the experimental variant and Control; notation: ***, but this is not in the tables of results. In tables, you use notation to denote that values are the average of three independent experiments ± standard deviations. The results of the experiment are completely but statistical analyses are showed in the table not clear. Please correct it.
  • Figure 7 (a, b, c) need to be changed (size). In this form, the figure is completely unreadable.

Author Response

Answers for Reviewer 2

We greatly appreciate your remarks on our manuscript and we would like to express our thanks. We made all the modifications requested by Reviewer.

please describe in detail the conditions in growth chamber (e.g. temperature, humidity and photoperiod),

Line 295-296. The specific conditions in growth chamber were added.

You specify that you are using the notation: *0.01 < p<0.001 = distinct significant differences between the experimental variant and Control; notation: ** p < 0.001 = highly significant differences between the experimental variant and Control; notation: ***, but this is not in the tables of results. In tables, you use notation to denote that values are the average of three independent experiments ± standard deviations. The results of the experiment are completely but statistical analyses are showed in the table not clear. Please correct it.

In order to have more explicit comments on plants growth parameters’, since statistical data are sometimes more difficult to be analyzed, we randomly selected 12 plants from all pots and the average of parameters values of these 12 independent plants are presented in Table 6. We consider that to a certain extent this presentation is more explicit and can be analyzed more clearly, this is the reason why we have introduced Table 6.In Fig 7. we presented our results as standard error bars and respectively as p values. For this statistical analysis, 50 plants (each plant growing in a tray) were evaluated for growth parameters. Tables 1-4 presented “values which are the average of three independent experiments ± standard deviations”. 

Figure 7 (a, b, c) need to be changed (size). In this form, the figure is completely unreadable.

We removed the unclear photos and added better ones.

Thank you for support and collaboration.

Yours sincerely

Luiza Jecu

17.02.2021
